# Effects of Freeze–Thaw Cycles on Soil Nitrogen Transformation in Improved Saline Soils from an Irrigated Area in Northeast China

Siyu Nie [1], Xian Jia [2], Yuanchun Zou [3] and Jianmin Bian [4,*]

1 School of Water Conservancy & Environment Engineering, Changchun Institute of Technology, Changchun 130012, China; nsy727@126.com
2 Songliao River Water Resources Commission of Ministry Water Resources, Changchun 130021, China; jiaxian727@163.com
3 State Key Laboratory of Black Soils Conservation and Utilization & Heilongjiang Xingkai Lake Wetland Ecosystem National Observation and Research Station & Key Laboratory of Wetland Ecology and Environment & Jilin Provincial Joint Key Laboratory of Changbai Mountain Wetlandand Ecology, Northeast Institute of Geography and Agroecology, Chinese Academy of Sciences, Changchun 130102, China; zouyc@iga.ac.cn
4 College of New Energy and Environment, Jilin University, Changchun 130021, China
* Correspondence: bianjm@jlu.edu.cn; Tel.: +86-180-0431-9968

**Abstract:** Freeze–thaw cycles (FTCs) occur during the nongrowing season, and residual nitrogen (N) increases the risk of N loss with melting water. To study the effect of FTCs on soil N, rice fields in improved irrigated saline soil in northeast China were selected as the research subjects. Water content (10%, 20%, and 30%), different N fertilizer levels (180 and 220 kg/ha), and multiple FTCs of soil samples were used to clarify the effects of N fertilizer application and water content on N efficiency. The results indicate that, after the third FTC, the soil ammonium nitrogen ($NH_4^+$-N) level increased significantly. $NH_4^+$-N increased with an increase in the initial soil moisture content and decreased with fertilizer levels. Nitrate nitrogen ($NO_3^-$-N) decreases with increasing initial soil moisture. The inorganic N increased significantly compared with that in the unfrozen stage, indicating that FTCs promote soil N mineralization. However, high fertilization rates inhibit mineralization. Analysis of variance showed that $NO_3^-$-N is sensitive to the N application rate, water content, and salinity ($p < 0.05$). FTCs and artificial fertilization are the factors that affect N mineralization ($p < 0.05$). The research results are significant for preventing nitrate leaching and soil acidification during spring plowing and providing a scientific basis for fertilization systems and water environment pollution in improved saline soils.

**Keywords:** soil nitrogen; freeze–thawing cycles; soil N mineralization; soil moisture and salinity; Northeast China

## 1. Introduction

Freeze–thaw cycles (FTCs) are a common phenomenon caused by seasonal or diurnal temperature variations, which are frequently encountered in higher-latitude and higher-altitude regions [1]. It changes the soil structure and physical and chemical properties through the solid–liquid phase transition of soil water and frequent FTCs [2]. This can seriously affect soil composition and nutrients. Global warming will significantly affect the freeze–thaw pattern in high-latitude and high-altitude permafrost areas and seasonal freeze–thaw areas. It is becoming increasingly important to understand the response of soil biogeochemical cycles to FTCs in terrestrial ecosystems.

FTCs can change soil physicochemical and biological characteristics by regulating soil physical structure and hydrothermal conditions, thereby affecting soil nitrogen (N) conversion to different degrees [3]. These changes in biogeochemical cycles may further

affect nutrient supply and crop growth and development in subsequent growing seasons. Transformation of soil N is vital for plant productivity and related ecosystem function and use because N is an essential nutrient for soil organisms [4]. FTCs occur mostly in the nongrowing season. Due to the time lag between soil nutrient supply and crop nutrient utilization, residual N in the soil after the growing period may increase the risk of soil nutrient loss [5–7]. Therefore, it is important to study the effect of soil freeze–thaw on water and N in agricultural irrigation areas.

Environmental changes will change FTCs, subsequently affecting the intensity and duration of soil frost and changing N mineralization and nitrification rates. Mineralization is the decomposition of organic N to ammonium ($NH_4^+$) and nitrate ($NO_3^-$). This is how plants mainly obtain the available N from the soil. This process changes the soil N dynamics [8]. For farmland irrigated areas, a continual increase in agricultural N fertilizer substantially increases the amount of total N and alters the rates of N cycling [9,10]. In addition, different initial soil conditions change soil physical and chemical properties under the influence of freezing and thawing. The effect on soil N mineralization is related to N loss and efficient utilization during spring tillage after FTCs [11].

Previous studies have observed that FTCs stimulate soil N migration and transformation and affect N conversion processes [12,13]. Song and Zhang [14,15] found that transitional zones disrupt microorganisms and soil aggregates and that additional nutrient availability stimulates the activity of the remaining microorganisms, increasing N mineralization. Increased frequency, intensity, and duration of FTCs are conducive to increasing soil inorganic N mass fraction and accelerating N mineralization rate [16,17]. FTC intensity can significantly increase soil nitrate nitrogen ($NO_3^-$-N) [18]. Some scholars have discussed the use of FTCs to promote N mineralization and improve the utilization rate. Fu [19] explored the effects of biochar application combined with soil moisture on N mineralization during FTCs to provide guidance for regulating farmland N in black soil with seasonally frozen soil. Because FTCs destroy soil aggregates [20], kill soil microorganisms, and damage plant roots, the content of ammonium nitrogen ($NH_4^+$-N) and $NO_3^-$-N in the soil [21] and the effect of FTCs on N depends on the freeze–thaw frequency. Herrmann conducted 20 FTCs in farmland soil within 40 days and found that with an increase in freeze–thaw frequency, the soil respiration and N mineralization rates gradually decreased [22]. Similar results have also been found in tundra and forest ecosystems and the black soil of farmlands.

The western part of Jilin province is a semiarid area and a typical seasonal frozen soil region with severe saline soil in northeast China. To improve land use efficiency, many paddy fields were developed in the area. Fertilization and irrigation changed the soil's physical and chemical properties. The newly added paddy fields increased agricultural nonpoint and pollution sources, affecting the migration and transformation behavior of N in soil and groundwater [23]. Although soil salinity has been reduced through irrigation, stopping soaking or improper fertilization may increase soil salinity and thus affect cultivation. These events alter the ecosystem's structure and function [24]. One study reported that an increase in low salinity can stimulate N mineralization and that different N types and applied fertilizer amounts affect the change in N [25]. It is extremely urgent to study the dynamic change of soil N and how it influences saline soil that develops and improves under freeze–thaw conditions. Currently, there are few studies on the effect of freeze–thaw on N conversion in seasonally frozen soil, which is of great significance to the soil and groundwater environment in improved saline–alkali soil.

Therefore, we chose improved saline soil as the research object because improved saline soil has a higher N application rate than nonsaline soil, and the variation characteristics of N are related to the soil nutrient composition during spring tillage after thawing, which can lay a foundation for ensuring stable growth of grain yield. The specific objectives of this study are as follows: (1) to study the changes in soil salt, water, and inorganic nitrogen contents during freeze–thaw cycles; and (2) to investigate the effects of different initial water content and fertilizer amount on nitrate, ammonium, and soil N mineralization under FTCs.

## 2. Materials and Methods

### 2.1. Site Description and Soil Sampling

Soil samples used in this study were collected from an irrigated area in an agricultural paddy field (123°52′48″ E, 45°18′45″ N), located in Da'an, Jilin Province, China, which experiences one of the most serious soil salinization problems, a continuous permafrost zone (Figure 1). Influenced by climate, topography, biology, and human activities, the soil in Da'an city is characterized by various types and complex distribution. There are seven main types of soil, including aeolian sandy soil, alluvial soil, marsh soil, chernozem soil, light chernozem soil, meadow soil, and saline–alkali soil.

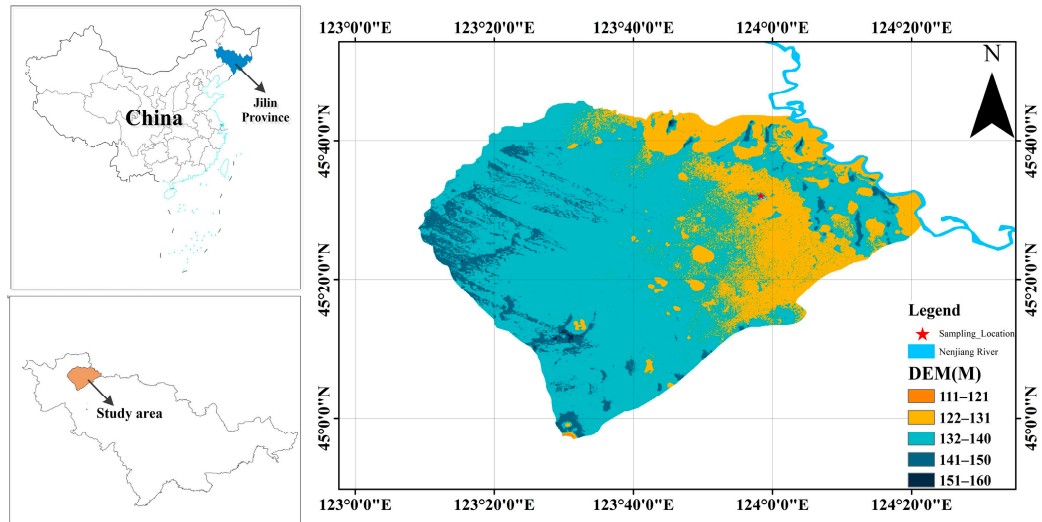

**Figure 1.** The location of study area.

The Da'an Irrigation Area project is built into a modern large-scale comprehensive agricultural irrigation area by diverting water from the Nenjiang River to the saline–alkali area of the Da'an ancient River channel, which raises moon bubble in the north and reaches Chagan Lake in the south. The annual average temperature is 5.1 °C from 1960 to 2020. The seasonal freezing period lasts for 5 to 6 months, with a depth of up to 1.7 m and a minimum temperature of −28 °C. The annual average precipitation is about 453 mm, mainly concentrated from June to August, and the annual average evaporation is about 1637 mm.

With the improvement of the saline soil and land consolidation project, there has been a large amount of saline–alkali land and wasteland in the paddy field. The type of N fertilizer used in this study area was all urea organic fertilizer with a 46% N content. An active layer of permafrost was present at soil depths of several meters in northeast China; therefore, the surface soil samples (0–30 cm) with different N fertilization (180 kg/ha and 220 kg/ha) at the site were collected using the multi-point mixing method in October 2020 and sieved (2 mm) to remove coarse roots and gravel. Visible roots were handpicked during sieving. Table 1 shows the soil physicochemical properties. Soil total nitrogen was detected using an automatic azotometer (Kjeltec 8400, FOSS, Copenhagen, Denmark) according to the Kjeldahl method. Soil pH was measured using a pH meter (SevenEasy, Mettler Toledo, Zurich, Switzerland). The soil moisture content was determined gravimetrically after drying at 105 °C [19]. The mechanical composition was measured using a laser particle size analyzer (Mastersizer3000, Malvern, United Kingdom).

**Table 1.** Soil physio-chemical properties of the sampling soil.

| Nitrogen Fertilization (kg/ha) | Soil Layer (cm) | Total Nitrogen (%) | pH | Bulk Density (g/cm³) | Soil Salinity (g/kg) | Mechanical Composition (%) | | |
|---|---|---|---|---|---|---|---|---|
| | | | | | | Clay | Silt | Sand |
| 180 | 0–10 | 0.075 | 8.8 | 1.29 | 1.76 | 38.8 | 30.3 | 30.9 |
| | 10–20 | 0.081 | 8.8 | 1.26 | 1.71 | 39.7 | 28.2 | 32.1 |
| | 20–30 | 0.087 | 8.9 | 1.32 | 1.83 | 38.9 | 26.7 | 34.4 |
| 220 | 0–10 | 0.131 | 8.9 | 1.34 | 1.82 | 38.6 | 23.7 | 37.7 |
| | 10–20 | 0.152 | 8.9 | 1.35 | 1.94 | 40.4 | 20.2 | 39.4 |
| | 20–30 | 0.164 | 9.1 | 1.55 | 2.12 | 43.7 | 20.5 | 35.8 |

*2.2. Test Device*

The laboratory freeze–thaw simulation test device is shown in Figure 2. Time domain reflectometry (TDR) multifunction sensors (LD-Y485, Shandong Lainde Intelligent Technology Co., Ltd., Shandong, China) were used to monitor soil water salinity, electrical conductivity, and temperature. The freeze–thaw simulation test equipment can provide a temperature range of −36.0 °C–60.0 °C (XT5405 series three-channel load freeze–thaw cycle test chamber, Hangzhou Xuezhongtan Constant Temperature Technology Co., Ltd., Hangzhou, China), which was connected to the temperature control plate at the bottom of the test box. Insulating foam material was used to ensure the freezing sequence of the soil column from top to bottom. The cylinder's sealing effect was good and effectively prevented the loss of water and nutrients. The soil column was a quartz glass column with a design specification of 40 cm in height and 10 cm in inner diameter. Holes with a height of 1.5 cm and a width of 4.5 cm were drilled into the left and right sides of the glass column from the top down 10 cm, 20 cm, and 30 cm, respectively. One side was used to insert the sensor probe, and the other side was used for sampling.

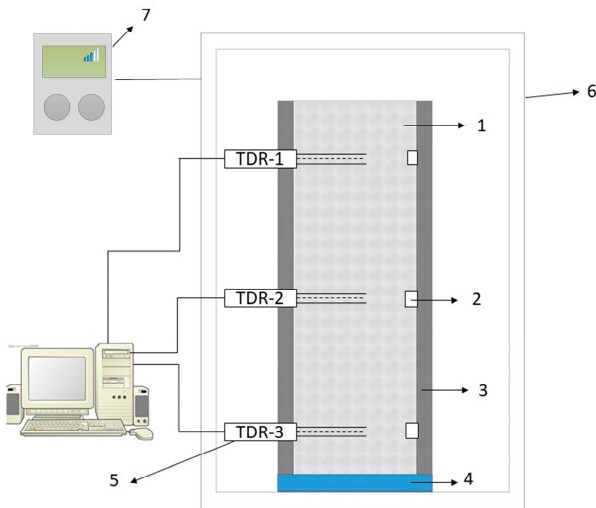

**Figure 2.** FTC device diagram (1—soil sampling; 2—sampling port; 3—foam insulation; 4—temperature control plate; 5—TDR; 6—freeze–thaw cycle test chamber; 7—temperature modulator).

*2.3. Experimental Design*

According to the fertilization habits in the study area, soil samples were collected under two fertilization levels (180 and 220 kg/ha; N1 and N2, respectively). The collected soil samples were air-dried, homogenized, and ground. The samples were returned to the laboratory in a sampling bag to avoid light. After natural air drying, crushing, impurity removal, sieving, and mixing, the homogenized soil samples (fresh soil) under different fertilization levels were moisturized with deionized water or distilled water to achieve 10% (W1), 20% (W2), and 30% (W3) moisture content (actual moisture content error is

±0.5%), with a total of six soil columns (N1W1, N1W2, N1W3, N2W1, N2W2, and N2W3). The surface of the completed soil columns was covered with plastic film to prevent water evaporation. It was then placed in a low-temperature environment of 4 °C for 24 h to stabilize its conditions. The initial $NO_3^-$-N and $NH_4^+$-N concentrations were the initial concentrations of the corresponding soil. During filling, uniform loading and compaction were observed. Each treatment sample was randomly selected from a length of 10 cm (L1), 20 cm (L2), and 30 cm (L3) after every FTC.

According to the average temperature in the study area and the current relevant research, a FTC temperature of −20 °C–5 °C was selected as the research temperature in this experiment. By setting the temperature, the temperature was gradually reduced to the target freezing temperature (−20 °C) within 6 h and maintained at this temperature for 12 h (freezing process). The temperature was then reset to increase the freezing chamber temperature to 5 °C for 12 h (thawing), representing a FTC. The samples were subjected to five different freezing–thawing frequencies (0, 3, 5, 10, and 20 times) [26].

*2.4. Analytical Methods and Statistical Analyses*

Immediately after the FTC, soil samples were collected from three replicates at each treatment to test for soil moisture, electrical conductivity (EC), $NH_4^+$-N, and $NO_3^-$-N. Soil EC was measured using a TDR sensor. Because of the significant correlation between EC and salt content, EC replaced salt content to describe the trend of change [27], which is expressed in ms/cm. Soil $NH_4^+$-N and $NO_3^-$-N were determined using the KCl extraction method. Soils were extracted with KCl solution (fresh soil:2 M KCl = 1:10, shaking for 1 h) and filtered using filter paper. Specifically, the extract of each sample was filtered through a filter paper and analyzed using a continuum flow auto-analyzer (AA3, Bran+Luebbe, Hamburg, Germany). The soil N mineralization rate (SNMR) was used to analyze N transformation. The formula is as follows [28]:

$$SNMR = \frac{\Delta C}{\Delta t} \tag{1}$$

where *C* is the concentration of the inorganic N ($NO_3^-$-N + $NH_4^+$-N); *t* is the day time.

Statistical Product Service Solutions was used for the mathematical statistical analysis, and the figures were obtained using Microsoft Excel 2010 for Windows. The analysis of variance (ANOVA) was used to describe the effects of the frequency of FTCs on soil salt content, $NO_3^-$-N, $NH_4^+$-N, and their interactions, which was used to perform an ANOVA at a significance level of 0.05. Following the ANOVA, Tukey's tests were used to discriminate significant multiple comparison differences among treatments. Tukey's pairwise comparison was used to identify and compare means that differed at probabilities of ≤0.05. The Pearson correlation test was used to examine the relationships between them.

## 3. Results

*3.1. Soil Physical Properties*

### 3.1.1. Soil Moisture Characteristics

The soil moisture contents of each soil layer under different treatment conditions during the FTCs are shown in Figure 3.

After FTCs, most of the treatments showed an increasing trend, but the soil moisture in the 30 cm soil layer differed (Figure 3). The moisture content of the 10 and 20 cm soil layers increased gradually as the FTCs increased. The 10 and 20 cm soil layers of N1W1 and N2W1 increased by 5.3–6.4% compared with the initial values, and the range of change was relatively large.

N1W3 and N2W3 in the 30 cm soil layer fluctuated and declined. The content was 1.84–3.05% lower than the initial value. The phenomenon of upward migration of water was obvious, which was perhaps driven by the gradient of soil water potential, leading to the migration of unfrozen water from the lower layer to the upper layer. The water content at 20 cm was maximum at the end of the FTCs.

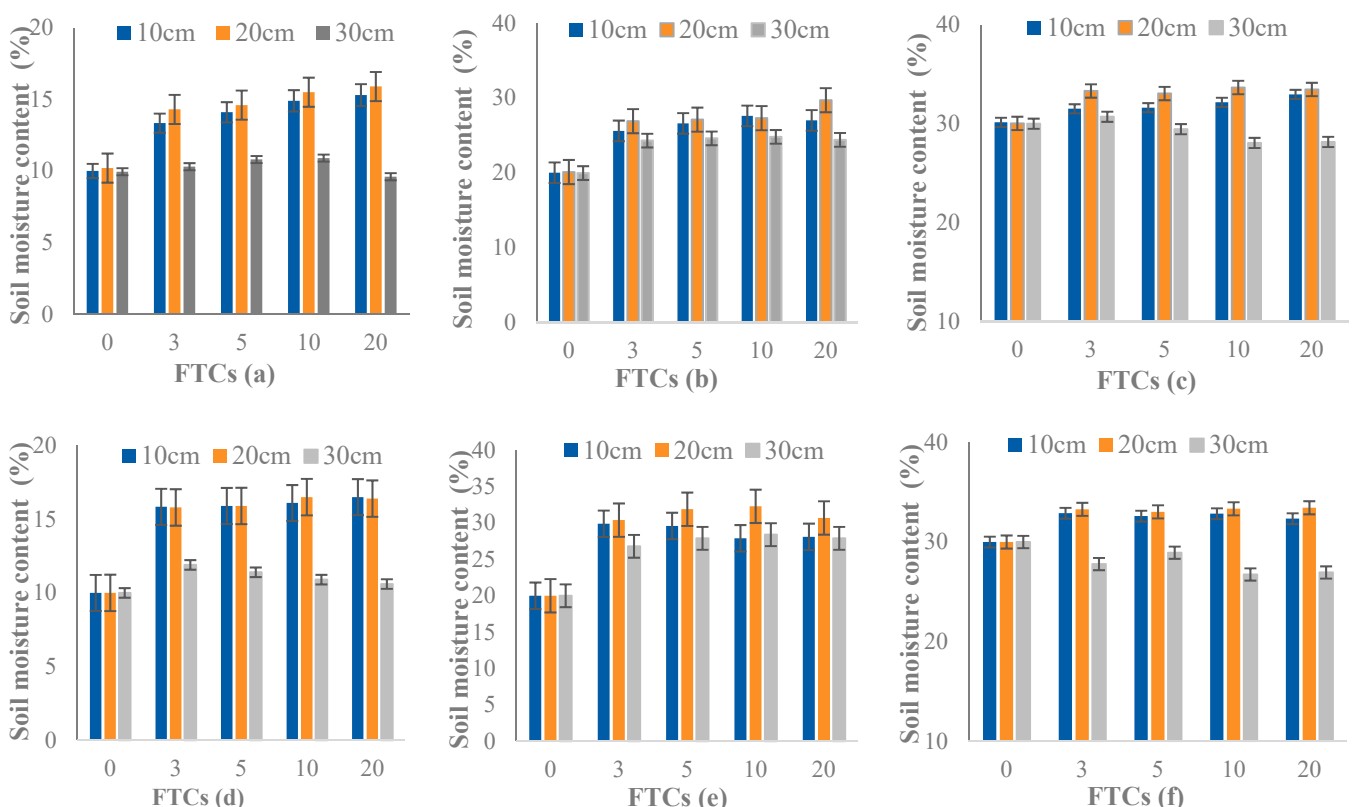

**Figure 3.** Changes in the moisture contents under the N1W1, N1W2, and N1W3 of 180 kg/ha (**a,b,c**, respectively), the N2W1, N2W2, and N2W3 of 220 kg/ha (**d,e,f**, respectively) during the FTCs in different soil layers.

With an increase in the N fertilizer application rate, the variation trend of the soil moisture content was consistent (Figure 3d–f). When the initial water content was small, the water content of the N2W1–N2W2 soil column increased by 0.5–1.2% and 0.6–3.5% after the third FTC, respectively, compared with that of the N1W1–N1W2 soil columns. For the group with a larger initial water content, the soil water content decreased by 0.5–1.2%. The increase in the initial water content and N application rate may have increased soil migration and soil water diffusion, but the difference was small. Due to the increase in the initial value, the entire soil moisture content increased after FTCs. Soil water migrates due to the action of the transition zone, unevenly distributing soil water, which eventually affects N mineralization.

It was observed that soil water was unevenly distributed and that the redistribution phenomenon was obvious. After FTCs, the moisture content at 20 cm in the soil column increased. This resulted from the downward migration of the upper layer and the upward migration of the lower layer. However, it changed less in the later cycles. Due to the high water content of the paddy field and the change from a moist area to a dry freeze–thaw test condition, water evaporation is large. It may also be caused by the transfer of water from the subsoil to the soil's surface.

### 3.1.2. Soil Salinity Characteristics

As shown in Figure 4, the salinity values of different soil layers treated with N1W1, N1W2, and N1W3 were 0.254, 0.314, and 0.451 ms/cm, 0.214, 0.397, and 0.492 ms/cm, and 0.411, 0.402, and 0.532 ms/cm, respectively.

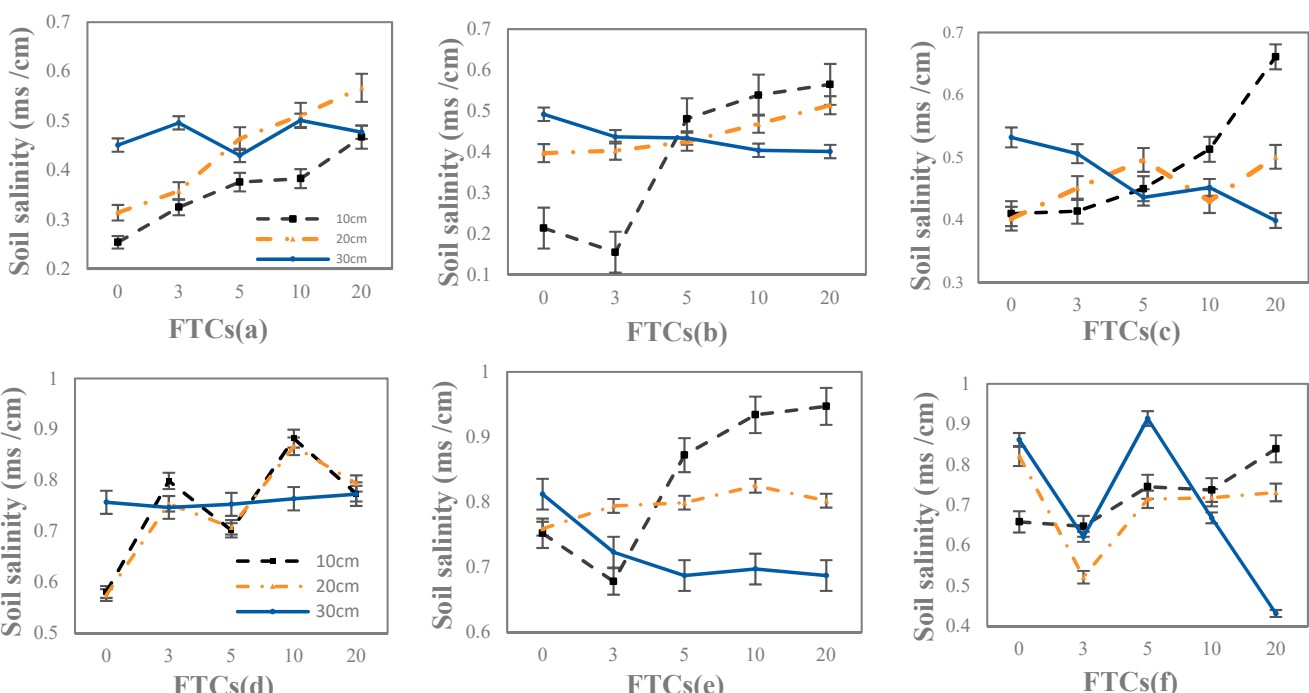

**Figure 4.** Changes in the electrical conductivity under the N1W1, N1W2, and N1W3 treatments of 180 kg/ha (**a**,**b**,**c**, respectively), the N2W1, N2W2, and N2W3 treatments of 220 kg/ha (**d**,**e**,**f**, respectively) during FTCs.

Compared with the initial values of N2W1, N2W2, and N2W3, the values decreased by 0.261–0.326, 0.32–0.52, and 0.248–0.419 ms/cm, respectively. These findings indicate that the initial soil salt content of the modified paddy field with a higher fertilizer application rate was higher. Furthermore, the salt content of the third layer of the paddy field was higher than that of the first and second layers. The findings were related to the soluble salt in water and infiltration with water. An increasing trend was observed after the third and fifth FTCs (Figure 4a–c). The 30 cm soil layer showed a downward trend after the fifth FTC, with increased initial water content.

With an increased fertilizer application rate, the salt content of the 10 and 20 cm soil columns of N2W1, N2W2, and N2W3 increased slightly. The final salt content exceeded 0.6 ms/cm (Figure 4d–f). After the FTCs, the soil salinity of the 30 cm soil layer in N2W2 and N2W3 significantly decreased, decreasing by 0.125 and 0.43 ms/cm, respectively. After the experiment, the water and salt contents in the upper soil column were higher than those in the lower soil column. This finding indicated that after FTCs, the upward accumulation of soil salt was significant with increasing water content when a large amount of fertilizer was applied. The change trend of salinity was consistent with a low fertilizer application rate.

### 3.2. Responses of the Nitrogen to FTCs

#### 3.2.1. $NH_4^+$-N Content

The variation characteristics of the soil $NH_4^+$-N content under different conditions after all FTCs are shown in Figure 5.

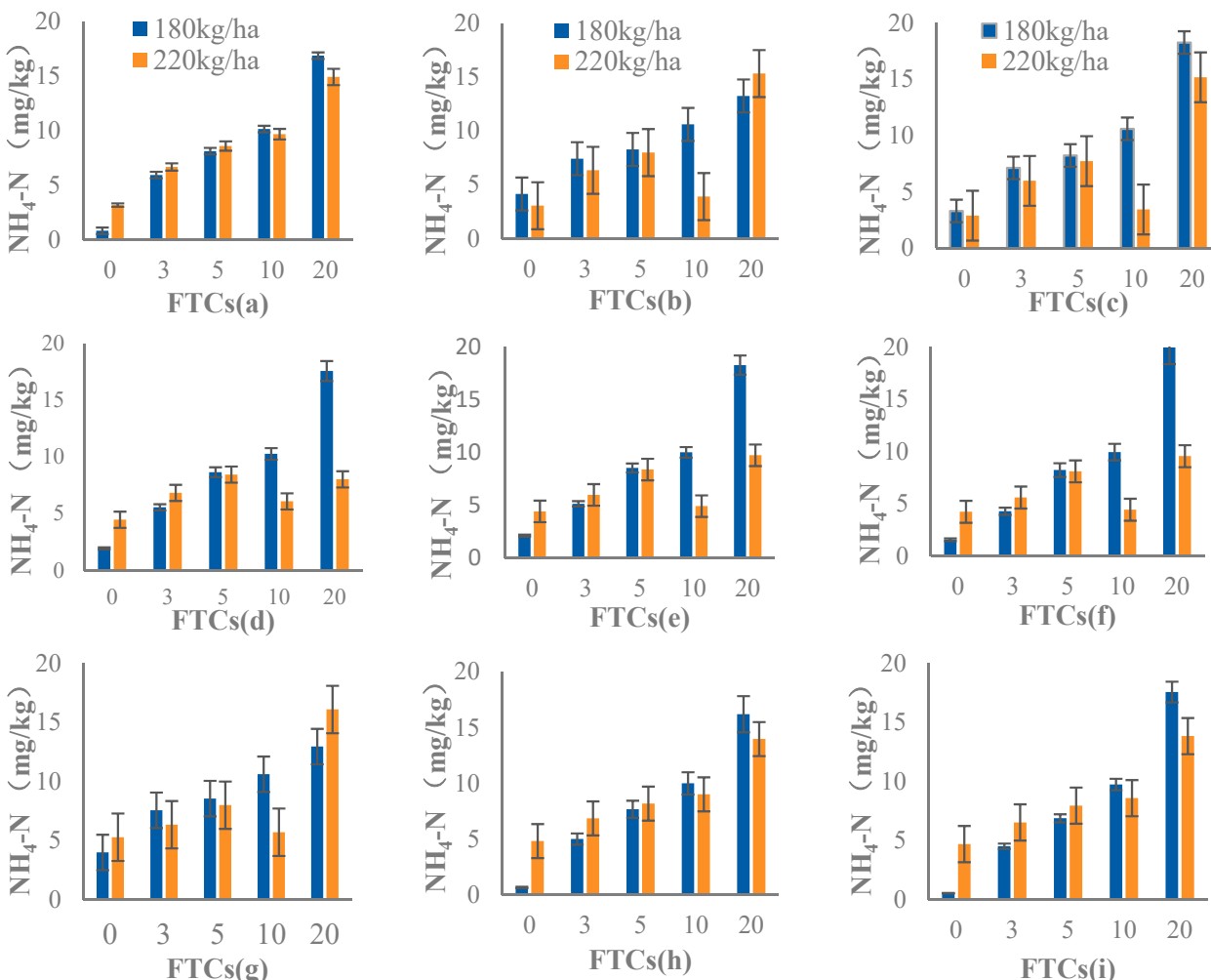

**Figure 5.** Variation characteristics in the NH$_4^+$-N contents of moisture content W1 (10%), W2 (20%), and W3 (30%) in the 10 cm soil layer (**a**–**c**), W1, W2, and W3 soils in the 20 cm soil layer (**d**–**f**) and W1, W2, and W3 soils in the 30 cm soil layer (**g**–**i**).

When the initial soil water content was 10%, the NH$_4^+$-N content in the 10 cm soil layer increased after the third FTC (Figure 5a–c). The NH$_4^+$-N content was higher than the initial level at the end of the FTC. However, as fertilizer application increased, the average growth range of NH$_4^+$-N decreased by 21–36% at different soil depths.

With increasing initial moisture content, NH$_4^+$-N content increased under FTCs (Figure 5c,f,i). The NH$_4^+$-N content of the W3N1 soil columns at different soil depths was 14.95, 18.40, and 17.04 mg/kg after a FTC, which was 15 times higher than that before a FTC. The NH$_4^+$-N content in the 20 cm soil layer was the highest. For the W3N2 soil columns, NH$_4^+$-N content decreased by 2.67, 13.08, and 7.89 mg/kg, respectively, compared with that of the W3N1 soil columns. Finally, the NH$_4^+$-N content increased with increasing initial soil water content, but the magnitudes of change differed.

### 3.2.2. NO$_3^-$-N Content

The variation in the characteristics of the soil NO$_3^-$-N content under different treatments after all FTCs is shown in Figure 6.

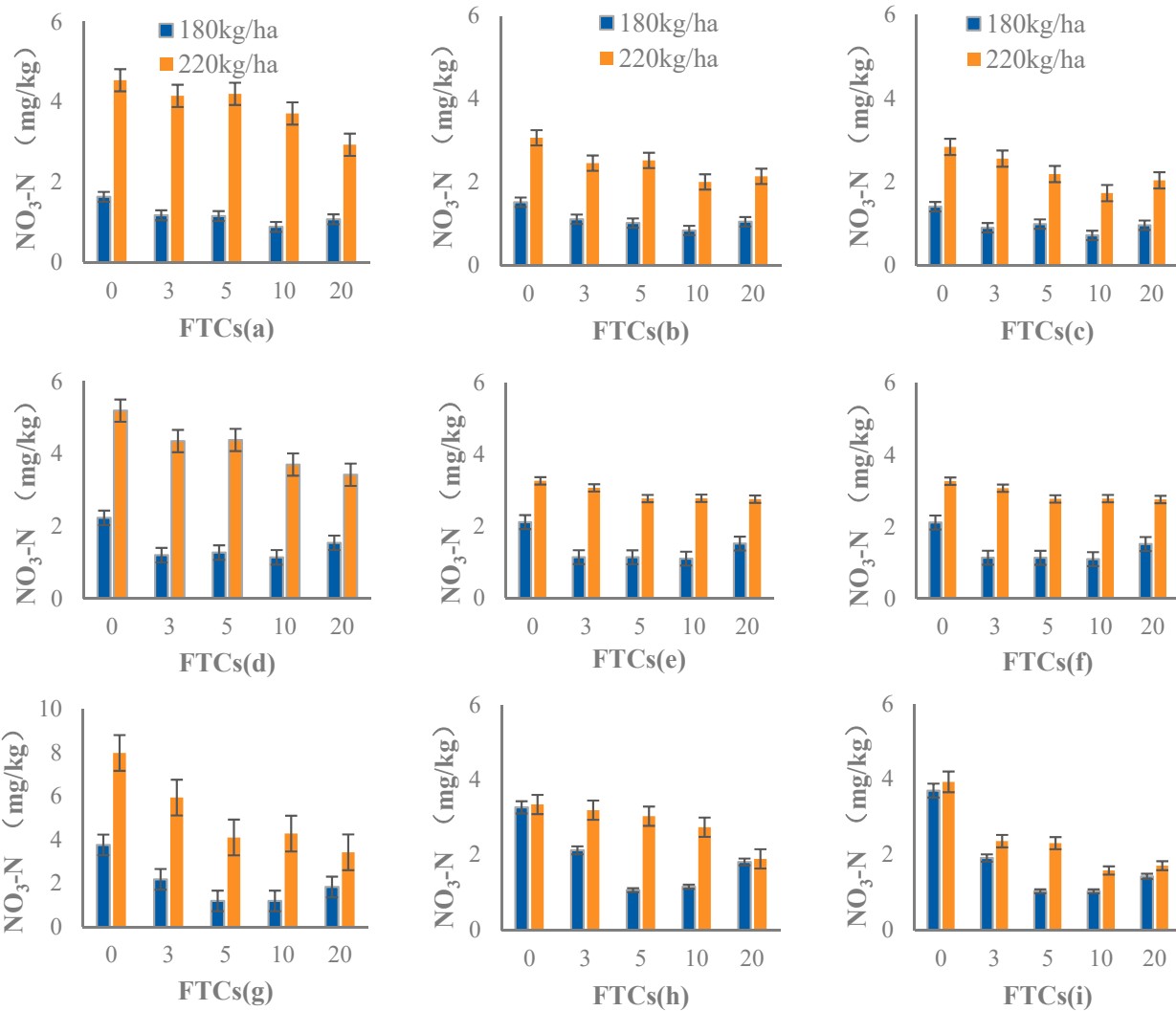

**Figure 6.** Variations in the $NO_3^-$-N contents of moisture content W1 (10%), W2 (20%), and W3 (30%) in the 10 cm soil layer (**a–c**), W1, W2, and W3 in the 20 cm soil layer (**d–f**) andW1, W2, and W3 in the 30 cm soil layer (**g–i**).

$NO_3^-$-N content decreased gradually with increasing FTCs at different N fertilizer levels (N1 = 180 kg/ha, N2 = 220 kg/ha). For the W1N1 soil column, $NO_3^-$-N content decreased gradually with soil depth. The respective content decreased by 34%, 30%, and 51% after 20 FTCs (Figure 6a,d,g). Figure 6g shows the greatest decline. For the W1N2 treatment, the decline was 35%, 34%, and 57% as the soil depth increased. As the N fertilizer application increased, the $NO_3^-$-N of the W1N2 soil column decreased by 1–6% more than that of the W1N1 soil column. Under the N2 treatment, the $NO_3^-$-N content varied greatly. As the initial water content increased, $NO_3^-$-N content gradually decreased, and the overall trend of change was consistent in different soil layers. According to Figure 6c,f,i, the reduction was 44–58% between the 20th FTC and the initial unfrozen soil. The $NO_3^-$-N content of the W3N2 soil column decreased by 3–30% more than that of the W3N1 soil column, and the most significant reduction was in the 20 cm soil layer. The results inferred that the $NO_3^-$-N content decreased significantly under the influence of FTCs when the soil moisture content was high.

### 3.3. Variation in the Characteristics of Nitrogen Mineralization under the Influence of FTCs

The mineralization rates of N were calculated by Equation (1), and results from the different treatment conditions during FTCs are shown in Figure 7.

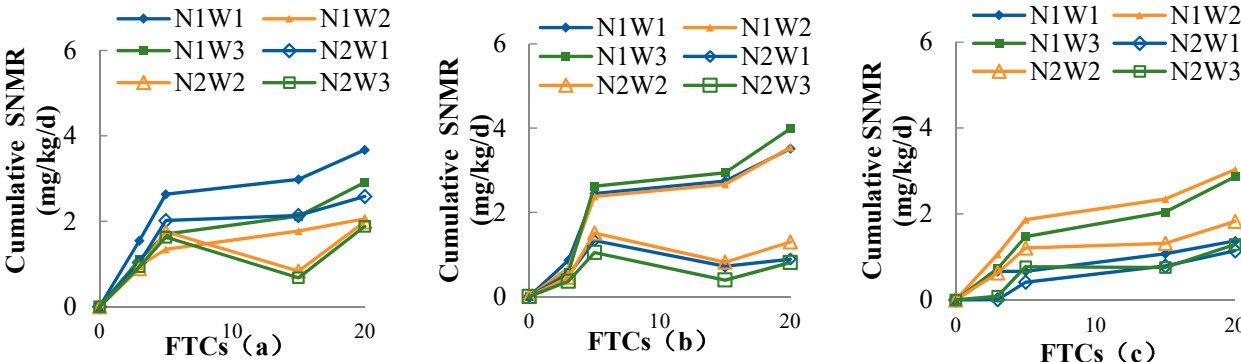

**Figure 7.** Changes in cumulative SNMR of N1W1, N1W2, N1W3, N2W1, N2W2, and N2W3 in 10 cm (**a**), 20 cm (**b**), and 30 cm (**c**) soil layers.

With the increase in FTCs, the inorganic N content of soil at different depths increased (Figure 7), and the maximum increase was 3.9 mg/kg/d at the end of FTCs. The cumulative SNMR of the surface soil (10 cm) was the highest, increasing by 2.5 mg/kg/d on average. It can be speculated that N mineralization in the surface soil is more active. The cumulative SNMR of surface soil (20 cm) under different water and N treatments had the greatest difference, ranging from 0.8 to 3.5 mg/kg/d, among which the cumulative SNMR under N2 treatment increased less. Before the fifth FTC, the cumulative SNMR tended to be steep from the early stage. Then, the change in cumulative SNMR significantly decreased at later times. An increasing freeze–thaw frequency could accelerate soil N to reach a relatively stable state, further increasing the soil $NH_4^+$-N content. Simultaneously, the SNMR decreased with increasing N fertilizer. However, in the 20 cm soil layer, the mineralization rate of the treatment groups with high N application rates (N2W1–N2W3) began to increase but significantly decreased after the fifth FTC, suggesting that increasing fertilization does not increase soil N mineralization much, but rather reduces the soil N mineralization rate.

Table 2 shows the significance level of the inorganic N and EC indices affected by different treatment conditions during FTCs. ANOVA showed that different treatments and their interactions significantly affected soil nitrate and ammonium contents ($p < 0.05$). FTCs and their interaction with fertilizer application rate significantly affected soil $NH_4^+$-N. The $NO_3^-$-N content was insignificantly affected by the interaction between different treatments, which was mainly affected by FTCs, moisture content, and N fertilizer. FTCs, N fertilizer, interaction between them, and interaction with the soil layer mainly influenced SNMR. N fertilizer affected EC. Moreover, Pearson's correlation analysis revealed that with 20 FTCs and different initial moisture content, soil EC concentration correlated significantly with $NO_3^-$-N concentration (R = 0.536; $p < 0.01$), and with no significant correlation with $NH_4^+$-N. $NO_3^-$-N and $NH_4^+$-N concentrations related negatively (R = −0.305, $p < 0.01$). Fertilizer application and its interaction with water content mainly affected salinity. The comprehensive analysis showed that the initial soil water content insignificantly affected the N mineralization rate.

**Table 2.** Results of analysis of variance testing the effects of FTCs, N fertilizer, moisture content, and soil layers.

| Source | $NH_4^+$-N | | $NO_3^-$-N | | SNMR | | EC | |
|---|---|---|---|---|---|---|---|---|
| | F-Value | Sig. | F-Value | Sig. | F-Value | Sig. | F-Value | Sig. |
| FTCs | **80.15** | **0.000** | **3.876** | **0.006** | **20.30** | **0.000** | 0.814 | 0.520 |
| Moisture content | 0.036 | 0.964 | **6.718** | **0.002** | 0.013 | 0.987 | 0.069 | 0.933 |
| Nitrogen fertilizer | 1.330 | 0.252 | **69.21** | **0.000** | **7.420** | **0.008** | **225.27** | **0.000** |
| Soil layers | 0.169 | 0.845 | 2.405 | 0.096 | 0.388 | 0.679 | 0.006 | 0.994 |
| FTCs × Moisture content | 0.317 | 0.957 | 0.177 | 0.993 | 0.503 | 0.850 | 0.310 | 0.960 |
| FTCs × Nitrogen fertilizer | **13.25** | **0.000** | 0.793 | 0.533 | **3.288** | **0.015** | 0.705 | 0.591 |
| FTCs × Soil layers | 0.379 | 0.928 | 0.515 | 0.841 | **4.096** | **0.000** | 0.846 | 0.565 |
| Moisture content × Nitrogen fertilizer | 0.051 | 0.950 | **11.97** | **0.000** | 0.176 | 0.839 | **3.314** | **0.041** |
| Moisture content × Soil layers | 0.009 | 1.000 | 0.188 | 0.944 | 0.456 | 0.768 | 0.230 | 0.921 |
| Soil layers × Nitrogen fertilizer | 0.566 | 0.570 | 0.101 | 0.904 | 1.265 | 0.288 | 1.506 | 0.228 |

Note: bolded numbers indicate statistically significant at $p < 0.05$.

## 4. Discussion

FTCs are an important driving force in the soil water and N cycle processes in cold ecosystems. Freezing and thawing of soil is a phased change process of soil moisture [28]. Driven by the gradient of soil water potential, the unfrozen water in the lower layer migrates to the upper layer, unevenly distributing the soil water. Water migration in soil drives changes in salt and nutrients in the soil. FTCs affect aggregate stability, which then affects soil N mineralization. The influence is mainly related to factors such as freezing and thawing patterns and soil properties, which then affect the changes in $NO_3^-$-N and $NH_4^+$-N content in the entire farmland soil [29]. Clarifying the internal mechanism of the special soil N cycle and transformation process under freezing and thawing conditions is important for future exploration of the quantitative characterization of the relationship between soil water and heat conditions and soil N migration and transformation capacity.

### 4.1. Effect of FTCs on the N Content

N is critical for crop growth. The initial water content and exogenous N change the content of $NH_4^+$-N and $NO_3^-$-N in the soil (Table 2). This experimental study showed that the effects of freezing and thawing conditions and their interactions on soil inorganic N are complex and unstable [12]. In this study, different levels of N fertilizer and soil water content significantly increased the $NH_4^+$-N concentration in all treatments [30,31]. The overall content was still higher than the initial level during FTCs (Figures 5 and 6). The inorganic N mass fraction in soil increased with the increase in FTCs, and a high frequency of FTCs promoted the accumulation of inorganic N in the soil [19,32]. This study is similar to the research results of Sulkava et al. (2003) [33,34]. They found that during the first FTC, the net N mineralization rate of soil was relatively fast, but with the increase in freeze–thaw frequency, the net N mineralization rate gradually decreased, indicating that fewer FTCs benefited soil N mineralization. The main reasons are as follows: (1) the residues of some dead microorganisms provide sufficient sources for remaining microbial activities, stimulating microbial activity and facilitating the mineralization process of soil organic N [35]; (2) FTCs alternation causes the expansion and contraction of soil pores, ultimately changing the soil crystal lattice structure and releasing fixed ammonium N [36]; and (3) FTCs cause more nutrients to be released, which may lead to biochemical reactions in surviving microorganisms and enhance nitrogen mineralization.

However, we observed that the cumulative SNMR tended to be stable, indicating that the mineralization rate in the later stage was weak (i.e., mineralization was inhibited). In addition, comparing the different soil depths revealed that the water content in the 20 cm layer was larger and the final mineralization rate was higher. Combined with the change in fertilizer application rate, excessive fertilization may inhibit N mineralization.

Herrmann [22] conducted 20 FTCs on farmland soil over 40 days and found that the rate of soil N mineralization gradually decreased with an increase in freeze–thaw frequency. The change rule of soil N mineralization in saline soil also matches the change in freeze–thaw frequency. The net N mineralization rate decreases with increasing freezing–thawing frequency (Figure 7). This may be due to two reasons. First, compared with field in situ experiments, indoor simulation experiments do not cover all factors, lacking N absorption and utilization by plant roots and the leaching loss of N due to precipitation, and the N mineralization rate is inhibited by mineralized N accumulated in the soil [37]. In addition, freezing and thawing coupled with salt effects reduce microbial activity and are inhibited [38]. This also affects the N mineralization rate. Consequently, the long-term effect of freezing and thawing on N mineralization is disturbed.

### 4.2. Effects of Soil Salinity and Exogenous Nitrogen on Nitrogen

The study area was highly saline before development. The maximum salt content of the land exceeded 6%. In recent years, many saline soils have been used as paddy fields. To increase rice yield, some farmers have increasingly applied fertilizer. The effects of freeze–thawing on N migration and transformation in saline soil remain unexplored. In recent years, many saline–alkali lands have been used for rice cultivation. To enhance rice yield, farmers increasingly employ fertilizers, thereby increasing the risk of exogenous nitrogen input and subsequent soil and groundwater pollution. However, the impact of artificial N input combined with freezing–thawing on N migration and transformation in saline soil remains unexplored. This study addressed this topic by conducting multiple freezing–thawing experiments using two types of soil with varying levels of N application in improved saline soil. The results revealed that under high-salt soil conditions, the $NH_4^+$-N content increased after FTC treatment while the $NO_3^-$-N content decreased.

Previous studies have indicated that winter maintains a high level of soil enzyme activity that is sensitive to FTCs [39]. Soil enzyme activity plays a crucial role in various N cycle processes such as nitric oxide reductase and nitrate reductase. $NH_4^+$-N is associated with N mineralization. However, excessive N levels may inhibit this process (Figure 7), which is particularly evident after three FTCs due to higher salt concentrations in the soil (Figure 4), leading to microbial inhibition [40]. Consequently, applying more N fertilizer does not increase the N mineralization rate or excessively elevate $NH_4^+$-N levels (Figure 5). The fluctuation trend observed for $NO_3^-$-N differs from most studies conducted on nonsaline soils (black soils) [41]. The possible reason for this phenomenon is that high-frequency soil FTCs change soil moisture from liquid to solid phase, which is fixed in soil pores, reducing the oxygen content in the soil, making soil anaerobic, inhibiting the activity of aerobic microorganisms, enhancing the activity of anaerobic microorganisms, weakening nitrification, and promoting denitrification. Therefore, under high-frequency FTCs, $NO_3^-$-N consumption increases, and the promotion effect of FTCs on nitrification is masked by denitrification. Under high levels of artificial N application, the phenomenon of surface salt accumulation in soil is evident, with high salt content (Figure 4). It was also found that EC was directly proportional to $NO_3^-$-N content (Pearson's correlation analysis). This may be due to the increased nitrification activity resulting from the application of more fertilizer, producing more hydrogen ions corresponding to soluble $NO_3^-$ to replace various cations [42], acidifying the soil.

Based on the principle that salt moves with water, it can be inferred that increasing the amount of exogenous nitrogen may increase the diffusion of soil moisture (Figure 3), and water flow will increase the risk of N loss. It is also possible that N in saline soil, such as ammonium salts, can react with alkaline components in the soil to produce gaseous N, which also causes N loss. Therefore, the water in the high-moisture treatment group in this experiment to some extent increases the activity of nitrate reductase and nitrite reductase, reducing $NO_3^-$-N to $NH_4^+$-N in the soil, thereby decreasing $NH_4^+$-N content in the soil while increasing $NH_4^+$-N content in the soil. Therefore, the risk and impact of soil nitrogen loss cannot be ignored. It is necessary to thoroughly consider the changes in microbial

activity in saline soil during freezing and thawing to reflect their relationship with soil mineralization capacity, thereby reducing N loss rates. In particular, irrigation of the study area before the spring plowing after the thawing period increases the risk of soil N loss and ecological environmental pollution. It will be the focus of research to coordinate spring plowing and improve N utilization efficiency in northeastern China.

**5. Conclusions**

The sensitive responses of $NO_3^-$-N and $NH_4^+$-N to FTCs in improved saline soils from an irrigated area were discussed with different water and N fertilizer treatments. The main conclusions are as follows:

(1) As the initial water content increased, the entire soil moisture increased after FTCs. FTCs affected the redistribution of soil water, which reached a maximum at the 20 cm soil layer. When N fertilizer was increased, the upward accumulation of soil salt was significant with high water content after FTCs.

(2) FTCs and their interaction with the fertilizer application rate significantly affected soil $NH_4^+$-N. FTCs can promote the increase in $NH_4^+$-N. However, as N fertilizer application increased, the average growth range of $NH_4^+$-N content decreased by 21–36%. $NO_3^-$-N content decreased gradually with increasing FTCs, which was mainly affected by FTCs, moisture content, and N fertilizer. Extensive fertilization increased the accumulation of soil $NO_3^-$-N. Salinity correlated positively with $NO_3^-$-N. $NO_3^-$-N in water-rich soil is sensitive to soil fertility and salinity.

(3) The inorganic N content of the soil at different depths increased, and the N mineralization rate in the later stage was weak. The soil N mineralization rates were enhanced as the FTCs increased but were inhibited at a high rate of fertilizer application. Applying more N fertilizer does not increase the N mineralization rate or excessively elevate $NH_4^+$-N levels. The fluctuation trend observed for $NO_3^-$-N differs from most studies conducted on nonsaline soils (black soils).

Through this study, we initially explored an interesting process of soil N after FTCs in this special improved saline soil. FTCs promoted soil N mineralization, but the effect on soil N retention needs to be verified through effective experiments in the following work. How nitrogen input changes the soil freeze-thaw pattern on N migration and transformation and the N biochemical cycle remains unclear. Further studies on the mechanism and effect of freeze–thaw changes on soil N conversion in saline soil under the background of temperature increase are required. It has an important effect on soil and groundwater pollution. This study's results can be used as a starting point to investigate the effects of FTCs on N morphology and salinity. It is necessary to quantitatively reveal the mechanism of soil enzyme response to salt in the future.

**Author Contributions:** Conceptualization, S.N.; methodology, X.J.; validation, Y.Z. and J.B.; formal analysis, S.N.; resources, J.B.; writing—original draft preparation, S.N.; writing—review and editing, Y.Z. and J.B. All authors have read and agreed to the published version of the manuscript.

**Funding:** This research was funded by the Department of Science and Technology of Jilin Province, grant number (YDZJ202301ZYTS216), the Department of Science and Technology of Jilin Province, grant number (20230508089RC), and the National Natural Science Foundation of China grant number (42272299).

**Data Availability Statement:** Data are contained within the article.

**Conflicts of Interest:** The authors declare no conflicts of interest.

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
