# Peer review of "Effects of Freeze–Thaw Cycles on Soil Nitrogen Transformation in Improved Saline Soils from an Irrigated Area in Northeast China"

_water, doi:10.3390/w16050653_

Round 1

Reviewer 1 Report

Comments and Suggestions for Authors

Editor

I have received and reviewed the manuscript titled “Effects of freeze-thaw cycles on soil nitrogen transformation in improved saline soils from an irrigated area in Northeast China”. This paper investigated the effects of freeze-thaw cycles on the nitrogen of soil samples with different water content and nitrogen fertilizer levels. The work is OK. there is innovation in the paper and the basic concepts and method of analysis are OK. However, I think a major revision is required. The following comments may be useful.

1. I'm not sure if it's a problem with the PDF file. All the figures in the manuscript I received are not clear. The author needs to double check them.

2. I suggest the author carefully revise the manuscript format according to the journal's requirements. For example, table 1, soil layer (cm). There are many similar problems in the manuscript. In addition, is g/kg a scientific unit for soil salinity?

3. When abbreviations first appear, The author should provide a detailed explanation of their meaning. Please carefully review the manuscript and revise the relevant content.

4. Lines 147 to 148 on Page 4. I suggest revising the descriptions to avoid confusion with subsequent descriptions (freeze-thaw cycle experiment). For example, “the freeze-thaw simulation test equipment can provide a temperature range of - 36.0 °C to 60.0 °C”.

5. I suggest that the author provide the design basic for the freeze-thaw cycle test, for example, the freezing and thawing temperatures of - 20.0 °C and 5.0 °C, the freezing time of 12 h and the thawing time of 12 h.

6. Lines 187 on Page 5, “Where” should be “where” when you explain the formula.

7. Lines 190 on Page 5, why Electrical Conductivity (EC) was used to replace the salt content for description? Please provide more information about EC.

8. I suggest the author provide some quantitative description for some interesting or significant changes. For example, Lines 277 on Page 8, the increase in NH4+-N content is most significant when the initiate moisture content is 30.0%. Please provide a quantitative description.

9. I suggest redrawing all data figures using a scientific drawing method. For example, Fig. 5 and Fig. 6, why is there no y-axis in the figure? The labels in the figure are also incorrect, (mg/kg). The author should carefully review and revise the manuscript to avoid similar problems.

10. Uniform significant digits should be used in the manuscript, especially for the same physical quantity. For example, Lines 225 on Page 6.

11. Lines 438 on Page 12, I suggest the author shows the conclusions of this study in the form of a list. After that, please make some suggestions and research ways for other researchers at the end of this section.

12. It is necessary to improve the language of the manuscript, some descriptions are unclear. For example, Lines 213 on Page 6, the author should provide a more clear and detailed description for Fig. 3. Similar problems need to be revised.

Comments on the Quality of English Language

It is necessary to improve the language of the manuscript, some descriptions are unclear. For example, Lines 213 on Page 6, the author should provide a more clear and detailed description for Fig. 3. Similar problems need to be revised.

Author Response

For research article

Response to Reviewer 1 Comments

1. Summary

Thank you very much for taking the time to review this manuscript. Those comments are all valuable and very helpful for revising and improving our paper, as well as the important guiding significance to our researches. Please find the detailed responses below and the corresponding revisions highlighted in the re-submitted files.

2. Questions for General Evaluation

Reviewer’s Evaluation

Response and Revisions

Does the introduction provide sufficient background and include all relevant references?

Yes

Thank you for your review.

Are all the cited references relevant to the research?

Yes

Thank you for your review.

Is the research design appropriate?

Yes

Thank you for your review.

Are the methods adequately described?

Can be improved

Thank you for your review. our corresponding response was given in the point-by-point response section.

Are the results clearly presented?

Can be improved

Thank you for your review. our corresponding response was given in the point-by-point response section.

Are the conclusions supported by the results?

Yes

Thank you for your review.

3. Point-by-point response to Comments and Suggestions for Authors

Comments 1: I'm not sure if it's a problem with the PDF file. All the figures in the manuscript I received are not clear. The author needs to double check them.

Response 1: Thank you for pointing this out. We have downloaded the PDF file from the system again for inspection. Figure 3-7 are all linked Excel, and I will update the figures in the paper. Thank you for your reminder.

Comments 2: I suggest the author carefully revise the manuscript format according to the journal's requirements. For example, table 1, soil layer (cm). There are many similar problems in the manuscript. In addition, is g/kg a scientific unit for soil salinity?

Response 2: It is really true as Reviewer suggested that we need to revise the manuscript format carefully. We neglected to bold the head of the tables. After examination, the problems were corrected. And soil salinity is usually expressed by mass concentration or volume concentration. scientific unit can be mg/g, g/kg, mg/L and g/m³, etc. According to the measurement results, g/kg is a scientific unit for soil salinity in this study.

Comments 3: When abbreviations first appear. The author should provide a detailed explanation of their meaning. Please carefully review the manuscript and revise the relevant content.

Response 3: Thank you for pointing this out. After reviewing and revising, we will pay more attention to this abbreviation problem. For example, Line 196 on Page 5. Thank you for your reminding.

Comments 4: Lines 147 to 148 on Page 4. I suggest revising the descriptions to avoid confusion with subsequent descriptions (freeze-thaw cycle experiment). For example, “the freeze-thaw simulation test equipment can provide a temperature range of - 36.0 °C to 60.0 °C”.

Response 4: We agree with your suggestion. Considering the suggestion, we have revised the sentence in Lines 147 to 148 on Page 4. The revised has been marked with highlight in red.

Comments 5: I suggest that the author provide the design basic for the freeze-thaw cycle test, for example, the freezing and thawing temperatures of - 20.0 °C and 5.0 °C, the freezing time of 12 h and the thawing time of 12 h.

Response 5: It is really true as Reviewer suggested that we need provide the design basic for the freeze-thaw cycle test. At present, in the research of seasonal soil freeze-thaw cycle, the freezing temperature is usually selected as -25℃ ~ -5℃, and the melting temperature is usually selected as 4℃ ~ 10℃. The average annual temperature in Da 'an City is 5.1℃. The previous field survey found that the soil temperature in the study area varied from -23.4 ℃ to 5.5 ℃ during the freeze-thaw period (November to March of the following year). Therefore, combined with domestic and foreign research and the meteorological characteristics of Da 'an City, -20℃ ~ 5℃ was selected as the freeze-thaw cycle temperature for this experiment. We have added explanations for changes in the Lines 164 on Page 5.

Comments 6: Lines 187 on Page 5, “Where” should be “where” when you explain the formula.

Response 6: Thank you for pointing out this issue. After checking, we have revised in Lines 187 on Page 5.

Comments 7: Lines 190 on Page 5, why Electrical Conductivity (EC) was used to replace the salt content for description? Please provide more information about EC

Response 7: We feel great thanks for your professional review work on our article. When it is not necessary to measure the ionic composition of the soil solution, the soil salinity sensor can be used to directly measure the conductivity of the soil solution to assess the soil salinity. The conductivity of soil solution EC is usually strongly correlated with soil salt content, and the conductivity can be used to study the trend of salt change, and the reference is added in the paper.

Comments 8: I suggest the author provide some quantitative description for some interesting or significant changes. For example, Lines 277 on Page 8, the increase in NH4+-N content is most significant when the initiate moisture content is 30.0%. Please provide a quantitative description.

Response 8: Thank you for pointing this out. Considering the suggestion, we have provided some quantitative description for some interesting or significant changes in Lines 277 on Page 8, Lines 293 on Page 9 and so on with highlight in red.

Comments 9: I suggest redrawing all data figures using a scientific drawing method. For example, Fig. 5 and Fig. 6, why is there no y-axis in the figure? The labels in the figure are also incorrect, (mg/kg). The author should carefully review and revise the manuscript to avoid similar problems.

Response 9: We are very sorry for our negligence of no y-axis in the figure. The ordinate is ammonium nitrogen and nitrate nitrogen in Fig. 5 and Fig. 6, its unit is mg/kg with no bold, and will correct it according to the standard of the journal. We also feel great thanks for point out.

Comments 10: Uniform significant digits should be used in the manuscript, especially for the same physical quantity. For example, Lines 225 on Page 6.

Response 10: We sincerely thank you for careful reading. The issue has been corrected highlight in red.

Comments 11: Lines 438 on Page 12, I suggest the author shows the conclusions of this study in the form of a list. After that, please make some suggestions and research ways for other researchers at the end of this section.

Response 11: We feel great thanks for your professional review work on our paper According to your nice suggestions, we rearrange the conclusions of this study in the form of list (1), (2), (3) with highlight in Conclusion. Moreover, the suggestions and unsolved problems are presented in the last paragraph.

Comments 12: It is necessary to improve the language of the manuscript, some descriptions are unclear. For example, Lines 213 on Page 6, the author should provide a more clear and detailed description for Fig. 3. Similar problems need to be revised.

Response 12: We are very sorry for our unclear description. Sorry for my poor English expression. We will try my best to revise and improve my English writing. Lines 213 on Page 6, the first sentence is intended to express that through the freeze-thaw cycle, most soil moisture shows an increasing trend, but soil moisture in the 30cm soil layer has a decreasing trend. There are different changing trends under different treatments. We have made corrections in the article and reviewed the full text to avoid Similar problem.

Special thanks to you for your good comments.

4. Response to Comments on the Quality of English Language

Point 1: It is necessary to improve the language of the manuscript, some descriptions are unclear. For example, Lines 213 on Page 6, the author should provide a more clear and detailed description for Fig. 3. Similar problems need to be revised.

Response 1: Thank you for your valuable and thoughtful comments. Questions about the unclear description of Figure 3 are answered in the Response 12, and we have carefully checked and improved the English writing in the revised manuscript.

5. Additional clarifications

[There are no any other clarifications you would like to provide to the journal editor/reviewer.]

Reviewer 2 Report

Comments and Suggestions for Authors

The article is devoted to the important issue of the repeated freeze-thaw impact on N cycling in irrigated agricultural soils. The authors have obtained a number of new data, а large pool of literature data on this topic has also been summarized. The objects chosen and the set of treatments also do not raise objections. However, the article still needs to be revised so that to clarify the following issues: how far the authors have progressed in accomplishing their tasks, which of the data obtained they believe critically important, what physical, chemical or/and microbiological mechanisms are to be considered to explain the results obtained etc. Below see the specific comments in more detail:

Comments

The forms of N applied as fertilizer (ammonium, nitrate, mixture of them, organic manure) should be described explicitly. For a while, it is not yet clear what kind of transformation i.e. denitrification, nitrification, ammonification, microbial immobilization physico-chemical immobilization with clay minerals etc - fertilizer nitrogen underwent before being involved in freeze-thaw cycles.

The authors analyzed a large body of literature on the effect of freezing-thawing on soil... therefore, it would be not difficult for them to formulate a working hypothesis that follows directly from the available literature data. Besides, the text of “Conclusions” (L.439 - 467) should contain information on how such a working hypothesis (suggestion, scenario) corresponds (or does not…) to the real data/results obtained.

I would recommend to the authors to make the main results of their research more specific. The sentences like …”Global warming will significantly affect the freeze-thaw pattern in high-latitude and high-altitude permafrost areas and seasonal freeze-thaw areas… At the same time, frequent agricultural activities change soil properties, resulting in a more complex and variable soil nitrogen conversion mechanism” seem pretty common and trivial, and do not contain any new information.

Also, it would be extremely useful to facilitate the main results as a “graphic result”; the latter would make it much easier for readers to understand the results of the research.

The use of the abbreviation NMR for N mineralization should be considered extremely confusing and misleading: this abbreviation is world-wide used to refer to the nuclear magnetic resonance technique.

Rename chapter ‘Discussions’ to ‘Discussion’

L. 102-103    Re-write the specific goals of the research: “The response factors of nitrogen transformation were revealed by ANOVA” definitely does NOT look like one of the goals.

The article can be recommended for publication after making appropriate changes to the text.

Comments on the Quality of English Language

English is OK.

Author Response

For research article

Response to Reviewer 2 Comments

1. Summary

Thank you very much for taking the time to review this manuscript. Those comments are all valuable and very helpful for revising and improving our paper, as well as the important guiding significance to our researches. Please find the detailed responses below and the corresponding revisions highlighted in the re-submitted files.

2. Questions for General Evaluation

Reviewer’s Evaluation

Response and Revisions

Does the introduction provide sufficient background and include all relevant references?

Must be improved

Thank you for your review. our corresponding response was given in the point-by-point response section.

Are all the cited references relevant to the research?

Yes

Thank you for your review.

Is the research design appropriate?

Yes

Thank you for your review.

Are the methods adequately described?

Yes

Thank you for your review.

Are the results clearly presented?

Must be improved

Thank you for your review. our corresponding response was given in the point-by-point response section.

Are the conclusions supported by the results?

Must be improved

Thank you for your review. our corresponding response was given in the point-by-point response section.

3. Point-by-point response to Comments and Suggestions for Authors

Comments 1: The forms of N applied as fertilizer (ammonium, nitrate, mixture of them, organic manure) should be described explicitly. For a while, it is not yet clear what kind of transformation i.e. denitrification, nitrification, ammonification, microbial immobilization physico-chemical immobilization with clay minerals etc - fertilizer nitrogen underwent before being involved in freeze-thaw cycles.

Response 1: Thank you for pointing this out. According to the survey statistics, the study area is an improved saline soil irrigation area, with an annual application amount of about 40,000 tons of nitrogen fertilizer, 9,000 tons of phosphate fertilizer and 33,000 tons of compound fertilizer. The utilization rate of nitrogen fertilizer in agricultural production is 30-35%. The nitrogen fertilizer in this study area is urea organic fertilizer with nitrogen content of 46%. We have added this information to Line 139 on page 4 with highlighted in red.

      It is really true as Reviewer suggested that not yet clear what kind of transformation underwent before being involved in freeze-thaw cycles. This is also one of the work contents of the subsequent growing season. Field monitoring was also carried out during the growing season. The data showed that the soil ammonium nitrogen content gradually decreased from the surface soil to the deep soil during the growing stage of rice, and the dynamic change of surface soil ammonium nitrogen was the most obvious during the growing stage of rice. The content of nitrate nitrogen in the upper layer (0-30cm) was higher, and the content of nitrate nitrogen decreased gradually along the soil profile(60-80cm). It is speculated that it is caused by the absorption and utilization of crop roots and the reaction of soil microorganisms. However, we only study their changes, focusing on the migration of water and nitrogen. The question of the transformation mechanism has not yet been thoroughly explored, and experimental studies on microorganisms in the growing season and non-growing season (freeze-thaw period) will be carried out soon. Thanks again for the comments of the reviewers, which also provide ideas for the future research.

Comments 2: The authors analyzed a large body of literature on the effect of freezing-thawing on soil... therefore, it would be not difficult for them to formulate a working hypothesis that follows directly from the available literature data. Besides, the text of “Conclusions” (L.439 - 467) should contain information on how such a working hypothesis (suggestion, scenario) corresponds (or does not…) to the real data/results obtained.

Response 2: Thank you for your comment. We analyzed the extensive literature on the effects of freezing and thawing on soil. The study area is a special high-salt area, and the amount of fertilizer applied after the improvement has always been large. The groundwater survey shows that the inorganic nitrogen content in the groundwater is also large. According to the characteristics of irrigation and fertilization in the study area, it is considered whether it is easier to leaching nitrate nitrogen if there is more water and the amount of fertilizer applied through freeze-thaw. Could the results be different from those in other regions? So, we designed this paper. Through this study, the law of nitrogen transformation was preliminarily explored, but the mechanism of nitrogen transformation was not taken into account when doing this experiment. In the subsequent work, the effects of microorganisms on nitrogen transformation have been considered. Meanwhile, the conclusions have been rewritten in accordance with the comments.

Comments 3: I would recommend to the authors to make the main results of their research more specific. The sentences like …”Global warming will significantly affect the freeze-thaw pattern in high-latitude and high-altitude permafrost areas and seasonal freeze-thaw areas… At the same time, frequent agricultural activities change soil properties, resulting in a more complex and variable soil nitrogen conversion mechanism” seem pretty common and trivial, and do not contain any new information.

Response 3: Thank you for pointing this out. We have deleted sentences that do not make scientific sense in the results after checking. The main result of the study was rewritten to be more specific according to the Reviewer’s suggestion.

Comments 4: Also, it would be extremely useful to facilitate the main results as a “graphic result”; the latter would make it much easier for readers to understand the results of the research.

Response 4: We agree with you and thank you for your comment. It is really true as Reviewer suggested that we need revise this part. We are sorry that we did not understand the display of "graphic result" in the result. We can understand that reviewer want us to collate the results, in order to facilitate the main results for readers to understand the results of the research. So, the main results are listed in order (1) (2) (3), and the last paragraph summarizes the work and problems that follow. Thanks again for your comment.

Comments 5: The use of the abbreviation NMR for N mineralization should be considered extremely confusing and misleading: this abbreviation is world-wide used to refer to the nuclear magnetic resonance technique.

Response 5: Thank you for pointing this out. We are very sorry for our incorrect writing. Considering the suggestion, we have revised the abbreviation, which change NMR to SNMR. SNMR represents soil nitrogen mineralization rate.

Comments 6: Rename chapter ‘Discussions’ to ‘Discussion’

Response 6: Thank you for pointing this out. Considering the suggestion, we have renamed chapter in Line 342 on Page 11.

Comments 7: L. 102-103    Re-write the specific goals of the research: “The response factors of nitrogen transformation were revealed by ANOVA” definitely does NOT look like one of the goals.

Response 7: Thank you for pointing this out. We agree that the specific goals of the research “The response factors of nitrogen transformation were revealed by ANOVA” definitely does NOT look like one of the goals. We deleted this sentence and integrated it with the others with highlighted in red.

Special thanks to you for your good comments.

4. Response to Comments on the Quality of English Language

Point 1: English is OK.

Response 1: Thank you for your review.

5. Additional clarifications

[There are no other clarifications you would like to provide to the journal editor/reviewer.]

Reviewer 3 Report

Comments and Suggestions for Authors

Figure 3 needs to be revised. The axis labels are not clear. if possible, please include color figures.

Conclusion section: Include the inferences from Table 2.

Table 2: Describe the numbers in bold.

Figure 4: Explain soil salinity unit 'ms/cm' in the figure caption or in the text.

Does FTC affect (a) the nitrogen uptake by plants? And (b)Nitrogen  immobilization?

Author Response

For research article

Response to Reviewer 3 Comments

1. Summary

Thank you very much for taking the time to review this manuscript. Those comments are all valuable and very helpful for revising and improving our paper, as well as the important guiding significance to our researches. Please find the detailed responses below and the corresponding revisions highlighted in the re-submitted files.

2. Questions for General Evaluation

Reviewer’s Evaluation

Response and Revisions

Does the introduction provide sufficient background and include all relevant references?

Yes

Thank you for your review.

Are all the cited references relevant to the research?

Yes

Thank you for your review.

Is the research design appropriate?

Yes

Thank you for your review.

Are the methods adequately described?

Yes

Thank you for your review.

Are the results clearly presented?

Yes

Thank you for your review.

Are the conclusions supported by the results?

Can be improved

Thank you for your review. our corresponding response was given in the point-by-point response section.

3. Point-by-point response to Comments and Suggestions for Authors

Comments 1: Figure 3 needs to be revised. The axis labels are not clear. if possible, please include color figures.

Response 1: Thank you for pointing this out. According to the suggestion, we have revised Figure 3, and replaced it with a color figure.

Comments 2: Conclusion section: Include the inferences from Table 2.

Response 2: We sincerely thank you for the valuable feedback that we have used to improve the quality of our manuscript. Considering the suggestion, Conclusions have been rewritten in accordance with the comments with highlighted in red.

Comments 3: Table 2: Describe the numbers in bold.

Response 3: Thank you for pointing this out. We are very sorry for our negligence. We have added footer to Table 2.

Comments 4: Figure 4: Explain soil salinity unit 'ms/cm' in the figure caption or in the text.

Response 4: We feel great thanks for your professional review work on our paper. The conductivity of soil solution EC is usually strongly correlated with soil salt content, and the conductivity can be used to study the trend of salt change, ms/cm or μS/cm is a unit of electrical conductivity in general. According to the suggestions, we have added the explain in the text with highlighted in red.

Comments 5: Does FTC affect (a) the nitrogen uptake by plants? And (b)Nitrogen immobilization?

Response 5: Thank you for your very good suggestion. Although these two problems were not considered in this paper, we have read relevant literature found that FTC could affect (a) the nitrogen uptake by plants and nitrogen immobilization. Some studies have shown that microbial nitrogen retention under snow cover and freeze-thaw is the main regulatory factor for nitrogen acquisition by vegetation during the growing season. The higher the microbial nitrogen fixation in freeze-thaw stage, the higher the plant nitrogen acquisition in growing season. In the follow-up monitoring work, we hope that in the future research on the quantification of the relationship between microorganisms and nitrogen transformation in saline soil, we will focus on to this issue.  It provides a good idea for the subsequent research. Thanks again for your comments.

Special thanks to you for your good comments.

4. Response to Comments on the Quality of English Language

[There is no Comments on the Quality of English Language]

5. Additional clarifications

[There are no other clarifications you would like to provide to the journal editor/reviewer.]

Round 2

Reviewer 2 Report

Comments and Suggestions for Authors

The authors have done a great job of correcting previously noted shortcomings, and overall it makes a good impression. However, it is surprising that after correcting the text, it was not proofread for basic errors/typos. The authors' English has definitely become much worse! In this connection, I would add the following comments:

1.    It is necessary to bring order to the use of singular and plural forms in relation to freezing/thawing processes. For example:

L 34 - Freeze-thaw cycles (FTCs) is a process – needs re-phrasing!

So, the text needs DEEP revision by a native English speaker. And again, one does not need to go far for examples (see below)!

2.    Some fragments have become poorly understandable.

LL 103 – 104 nitrate nitrogen, ammonium nitrogen and soil nitrogen mineralization – re-phrase e.g. to nitrate, ammonium and soil N mineralization or something like that.

L 292 - which the change is greatest at the 20cm layer - ????... what does it mean at all?

L 314 - high nitrogen application rates are not conducive to nitrogen mineralization - this sentence is hardly understandable.

L 365 - thereby enhancing the nitrogen mineralization of surviving microorganisms – this sentence is hardly understandable.

L 452 - Mineralization rates of nitrogenrates of N mineralization?

3.    Other comments

L 103 – remove and discuss

LL 183 – 185 - unfortunately, I missed the following note last time (sorry for that!) It is necessary to provide a detailed description of the method, with references to literary sources, the exact specification of an equipment, names of the manufacturer and the country of affiliation. Information about manufacturing companies and countries of affiliation should also be included in the description of other methods (this is a general rule). In general, it is necessary to provide such detailed information about the experiment that someone else can re-establish the entire experiment from beginning to end.

After such a revision, the article can be accepted for publication.

Comments on the Quality of English Language

The manuscript needs DEEP revision by a native English speaker.

Author Response

For research article

Response to Reviewer 2 Comments

1. Summary

Thank you very much for taking the time to review this manuscript. Those comments are all valuable and very helpful for revising and improving our paper, as well as the important guiding significance to our researches. Please find the detailed responses below and the corresponding revisions highlighted in the re-submitted files.

2. Questions for General Evaluation

Reviewer’s Evaluation

Response and Revisions

Does the introduction provide sufficient background and include all relevant references?

Yes

Thank you for your review.

Are all the cited references relevant to the research?

Yes

Thank you for your review.

Is the research design appropriate?

Yes

Thank you for your review.

Are the methods adequately described?

Can be improved

Thank you for your review. our corresponding response was given in the point-by-point response section.

Are the results clearly presented?

Can be improved

Thank you for your review. our corresponding response was given in the point-by-point response section.

Are the conclusions supported by the results?

Can be improved

Thank you for your review. our corresponding response was given in the point-by-point response section.

3. Point-by-point response to Comments and Suggestions for Authors

Comments 1: It is necessary to bring order to the use of singular and plural forms in relation to freezing/thawing processes. For example: L 34 - Freeze-thaw cycles (FTCs) is a process – needs re-phrasing! So, the text needs DEEP revision by a native English speaker. And again, one does not need to go far for examples (see below)!

Response 1: Thank you for pointing this out. We apologize for the poor language of our manuscript. We worked on the manuscript for a long time and the repeated addition and removal of sentences and sections obviously led to poor readability. We have now worked on both language and readability and have also involved native English speakers for language corrections. At the same time, we found an English editing service to help polish our article.

The meaning of this sentence in L 34 is that freeze-thaw cycles is common phenomenon and often occur at the soil surface and at certain depths below the surface, caused by seasonal or diurnal temperature variations. Reviewed the latest references on freeze-thaw cycles. Generally speaking, the freeze-thaw cycles is a phenomenon that occurs repeatedly, and the freeze-thaw cycles is abbreviated as FTCs, which is in the plural form. Some use the singular form when introducing the concept of the freeze-thaw cycle. In this paper, the number of freeze-thaw cycles was carried out many times during the experiment, so the plural form (FTCs) was used. We unify the form.

Comments 2: Some fragments have become poorly understandable.

LL 103 – 104 nitrate nitrogen, ammonium nitrogen and soil nitrogen mineralization – re-phrase e.g. to nitrate, ammonium and soil N mineralization or something like that.

L 292 - which the change is greatest at the 20cm layer - ????... what does it mean at all?

L 314 - high nitrogen application rates are not conducive to nitrogen mineralization - this sentence is hardly understandable.

L 365 - thereby enhancing the nitrogen mineralization of surviving microorganisms – this sentence is hardly understandable.

L 452 - Mineralization rates of nitrogen – rates of N mineralization?

Response 2: Thanks for your comment. We apologize for the poor language. According the suggestion, LL 103 – 104 was re-phrased with highlight in red.

L 292 -- which the change is greatest at the 20cm layer. The meaning of this sentence is that the NO3-N of W3N2 soil column is reduced by 3% to 30% compared to W3N1 soil column. The reduction of NO3−-N in the W3N2 soil column was most significant in the 20cm soil layer.

L 314 - high nitrogen application rates are not conducive to nitrogen mineralization. This means is that increasing fertilization does not increase soil nitrogen mineralization much, but rather reduces the soil N mineralization rate.

L 365 - thereby enhancing the nitrogen mineralization of surviving microorganisms. The meaning of this sentence is, FTCs release more nutrients, which may lead to biochemical reactions in surviving microorganisms and enhance nitrogen mineralization.

   L 452 - Mineralization rates of nitrogen. Thank you for pointing this out. We were really sorry for our careless mistakes. Similar issues have been corrected.

Comments 3: (1)L 103 – remove and discuss

(2) LL 183 – 185 - unfortunately, I missed the following note last time (sorry for that!) It is necessary to provide a detailed description of the method, with references to literary sources, the exact specification of an equipment, names of the manufacturer and the country of affiliation. Information about manufacturing companies and countries of affiliation should also be included in the description of other methods (this is a general rule). In general, it is necessary to provide such detailed information about the experiment that someone else can re-establish the entire experiment from beginning to end.

Response 3: (1) Thank you for pointing this out. We sincerely thank the reviewer for careful reading. We have removed the words. (2) We think this is an excellent suggestion. We have provided a detailed description of the method, with references to literary sources, the exact specification of an equipment, names of the manufacturer and the country of affiliation. For example, line 134-138, equipment for testing total soil nitrogen with highlight in red. And, in section “2.3 Experimental design”, we have added detailed information about the experiment. Since it is the first time to write an experimental article, there are many information that need to be paid attention to. We have gained a lot from this revision, which has helped us improve the quality of the article.

Special thanks to you for your good comments.

4. Response to Comments on the Quality of English Language

Point 1: The manuscript needs DEEP revision by a native English speaker.

Response 1: Thanks for your suggestion. We feel sorry for our poor writings and making the language harder to understand after the first revision. We tried our best to polish the language in the revised manuscript. For this revision, we sought help from teachers who speak English well and found an English editing service to polish our article, hoping to reach the standard and better understanding.

5. Additional clarifications

[There are no other clarifications you would like to provide to the journal editor/reviewer.]
